# Individual, Peer, and Family Correlates of Depressive Symptoms among College Students in Hong Kong

**DOI:** 10.3390/ijerph20054304

**Published:** 2023-02-28

**Authors:** Nelson W. Y. Tam, Sylvia Y. C. L. Kwok, Minmin Gu

**Affiliations:** 1Department of Social and Behavioral Science, The City University of Hong Kong, Kowloon, Hong Kong 518057, China; 2Research Institute of Social Development, Southwestern University of Finance and Economics, Chengdu 610074, China

**Keywords:** ecological model, individual, peer, and family correlates, college students, depressive symptoms

## Abstract

In this study, an ecological model and developmental psychopathology theory focusing on an ontogenic system (hopelessness) and microsystems (peer alienation and childhood abuse and trauma) was adopted to examine the individual, peer, and family correlates of depressive symptoms among college Chinese students in Hong Kong, China. A cross-sectional survey research design with a convenience sampling procedure was used to examine a sample of college students (*n* = 786) aged 18 to 21 years old in Hong Kong. Among them, 352 respondents (44.8%) reported having depressive symptoms, with a Beck Depression Inventory-II (BDI-II) score of 14 or above. The results of this study indicated that childhood abuse and trauma, peer alienation, and hopelessness were positively related to depressive symptoms. The underlying arguments and implications were discussed. The study results provided further support for the ecological model and the developmental psychopathology theory on the predictive roles of individual, peer, and family correlates of adolescent depression.

## 1. Current Situation of Depression in Adolescents in Hong Kong

The prevalence of depression in adolescents is increasing worldwide. Scholars have been concerned that depression in adolescents appears to be a worsening trend in Hong Kong [1,2,3]. Moreover, a representative study showed that a total of 9518 secondary school students, (38.36% males and 46.13% females), had a moderate to severe level of depression (The Center for Epidemiological Studies-Depression (CES-D) ≥ 21) [4]. Furthermore, depression was more prevalent among adolescents in Hong Kong compared to Macau and mainland China, further indicating its impact on the quality of life of adolescents [5].

### 1.1. Individual, Family, and Peer Risk Correlates of Depression among Chinese Adolescents

Research has shown that some individual, family, and peer correlates are positively associated with depression in Hong Kong. Concerning the influence of individual factors, constant, or deteriorating behavioral problems and low self-esteem [6], body dissatisfaction [7] and internet use-related addiction problems [8,9] are positively related to depressive symptoms among Chinese adolescents. In addition to the influence of family factors, parent-child triangulation [10], childhood neglect and parental suicidal ideation [11], and high parental expectations [12] are positively associated with depressive symptoms in Chinese adolescents. Furthermore, peer level, peer victimization [13], bullied victimization [14], and preference- for solitude (social withdrawal) [15] can increase the likelihood of depression symptoms in Chinese adolescents.

### 1.2. Impact of Depression

Depression may become a significant health problem if it is left untreated. Previous studies have shown that depressed individuals are likely to experience depressed moods, loss of interest or pleasure, change in appetite, change in sleep pattern, psychomotor impairment, low energy levels, feelings of worthlessness, lack of concentration, difficulties in making decision, and suicidal ideation [16,17]. These symptoms might lead to significant distress, a deterioration in social and occupational functioning [16], and disruption in the regular performance of typical roles [18]. Therefore, the depression can be so severe that it would impair one’s “occupational, academic, and most relevantly, social lives” [19]. 

Furthermore, adolescent depression can bring stress to parents and families [20]. Depressed individuals may increase the daily stress levels of their families, affecting parental relationships and family cohesion [4]. Therefore, there is an urgent need to better understand what contributes to adolescent depression in Hong Kong.

### 1.3. Conceptual Framework of the Study

The objective of this study was to investigate the personal, family, and peer correlates of depressive symptoms among adolescents in Hong Kong. Specifically, this study investigated the contributions of hopelessness, childhood abuse and neglect, and peer alienation to depressive symptoms in samples of adolescents in Hong Kong.

The selection of these factors was mainly based on different theoretical perspectives. First, this study adopts the ecological model of human development as its conceptual framework [21]. According to this model, human behavior could be interpreted by considering mutual interactions among individuals, as well as the social and cultural environments. The five ecological systems, including the ontogenic system, microsystems, the mesosystem, the exosystem, and the macrosystem interact with and influence each other.

According to the ecological model, the individual (ontogenic) system is the innermost level, which interacts with objects and familiar persons, as well as the current environment. An individual’s microsystem directly affects their development through parents, siblings, peers, and teachers, depending on the age of the individual.

Based on the framework of this model, many factors, such as peers, family, and religions are associated with the development of depressive symptomatology [22]. However, scholars have argued that ecological model focuses too much on the environmental and contextual (mesosystems, exosystems) influences on adolescent development [21,23]. In other words, the emphasis on the influence of the ontogenic system on adolescent development may be given less attention. Microsystems (e.g., peers and family) and current processes are particularly important in adolescent development studies [24]. In this study, demographic data is related to the exosystem, and if the current model includes too many factors from the mesosystem and exosystem, explanation of adolescent depression may be superficial and not concise enough. Therefore, an ecological model with a focus on the ontogenic system and the microsystem is applied to the study of adolescent depression.

Moreover, consistent with developmental psychopathology theory [25], this study intends to explain adolescent psychopathology problems (e.g., depression) from the perspective of human development. This perspective supports abnormal development in a dynamic relationship between the person and the environment. To better understand the developmental changes that take place in adolescents, it is better to understand the individuals, their environment, and their interactions. Therefore, based on the abovementioned theories, the ontogenic system and the microsystem are applied to the investigation of adolescent depression in this study.

### 1.4. Childhood Abuse and Trauma and Depressive Symptoms

Researchers generally agree that family factors occupy an essential role in developing, maintaining, and leading adolescent depression [26,27,28]. For instance, a study by Restifo and Bögels [27] concluded that family risk factors for youth depression consisted of four system levels, including a parent-child subsystem, a marital subsystem, a whole family subsystem, and an extrafamilial subsystem. Each subsystem can be used to identify predictors for depression and depressive symptoms among adolescents. Furthermore, Nydegger [29] pointed out that the whole family environment is likely to be an essential factor in the later development of depression in children and adolescents. Specifically, research has determined that childhood abuse increases the rate of depression in adolescents [30,31]. Theoretical frameworks, including attachment theory and cognitive behavioral theory, have been used to connect the association between the history of childhood abuse and depression in adolescents.

Attachment theory [32] holds that caregivers are essential to developing a child’s representations or internal working models for later life. Abusive experiences in childhood may greatly influence an individual’s schemas or thought patterns, which may transform to negatively impact current relationships. The negative experience of childhood abuse induces negative belief systems in individuals, including negative thoughts of themselves, the world, and their future, which eventually result in depression [33].

Furthermore, according to cognitive behavioral theory [34], childhood abuse leads to dysfunctional attitudes, resulting in ‘negative cognitive bias and increased susceptibility to depression when [schemas are] activated by stressful events’ [34]. Automatic thoughts are also related to stressful conditions that may lead to depression [35]. Moreover, empirical studies have found that automatic negative thoughts are related to depression [36,37].

On the other hand, Krug, Mercy, Dahlberg and Zwi [38] stated that cultural factors are particularly important in assessing child maltreatment because the cultural context can impact every aspect of life, such as aspects of human behavior (e.g., childrearing). Consequently, people in Hong Kong may present challenges regarding the understanding of child maltreatment. A Hong Kong study stated that traditional Chinese values such as filial piety of Confucius’ teachings impacts the Chinese view on understanding child maltreatment; the respondents of the study reported that they were prone to keeping childhood maltreatment history a secret because their loyalty to parents comes before anything else [39]. In the Chinese cultural context, the Chinese are trained to be group-oriented, relation-oriented, and socially interdependent by the teachings of Confucius. They may pay less attention to their individuality. Hence, the effect of childhood abuse and neglect on adolescent depression may be different. Therefore, it is crucial to study the relationship between child abuse and trauma and depressive symptoms in Chinese adolescents.

### 1.5. Peer Alienation and Depressive Symptoms

During adolescence, adolescents begin exploring wider social environments. They shift their attention from family to peer groups and relationships in school and community contexts [40]. The external environment (peer factors) is highly significant to their development [41,42]. Recent studies have found that peer groups have a role in the etiology of depression among adolescents [43,44]. According to Bronfenbrenner [21], alienation is a form of disconnection in which an individual is isolated from others in their social surroundings. According to Rayce, Holstein and Kreiner [45], alienation is a risk factor during adolescence because the feelings of ‘not belonging’ may create problems in interpersonal relationships and social exclusion, which in turn leads to several physical and mental health problems [45]. Moreover, adolescents who report feelings of ‘not belonging’ are those who find it challenging to make friends, which in turn contribute to a weak supportive network, and it is easy for them to have low self-esteem. These feelings are related to depression and other psychological problems [46].

Armsden and Greenberg [47] defined peer alienation as ‘feelings of isolation, anger, and detachment experienced in attachment relationships with peers’ [47]. A community study showed that feeling alienated from one’s peers corresponded to other problems experienced by adolescents and adults, including depression [48]. Another study found that perceptions of peer alienation were positively associated with anxiety sensitivity symptoms [49], which in turn increase the risk of secondary depression [50]. In a meta-analysis, Gorrese [51] reported that peer alienation was ‘significantly and positively associated with depression’ [51].

To the best of our knowledge, no research has been done to study the relationship between peer alienation and depressive symptoms among Chinese adolescents. Hence, it is worth studying the development of peer alienation in relation to depressive symptoms in Chinese adolescents.

### 1.6. Hopelessness and Depressive Symptoms

Based on the hopelessness theory of depression [52], when individuals who show a cognitive vulnerability encounter a stressful life event, their risk of developing depressive symptoms and becoming depressed increases.

Previous research has proposed using the hopelessness theory of depression to explain the onset of depression in mid- to late adolescence because some etiological factors, such as cognitive vulnerability and hopelessness, are developmentally operative [53]. Adolescents encounter many kinds of adverse life events, particularly adverse interpersonal events [54], that contribute to hopelessness.

Furthermore, studies have also found that hopelessness can predict the severity of future depression [55]. All of the above studies indicate that hopelessness and depression are strongly correlated.

However, according to one Hong Kong study, the Chinese are regarded as more optimistic than Westerners, believing that all things are cyclical, and hope always exists [17]. People who are flexible and follow nature’s flow always solve unsolvable problems [56]. An old Chinese proverb goes: “let us cross the bridge when we come to it” (Chuan dao qiao tou zi ran zhi). Therefore, the Chinese are likely to believe in having a positive future, even when they encounter negative issues in life. Hence, since Chinese culture emphasizes optimism and hope for the future, hopelessness may be considered to be less valuable by adolescents, which may result in fewer depressive symptoms. In short, hopelessness and depression are two separate constructs. Therefore, it is essential to examine the relationship between hopelessness and depressive symptoms in Chinese adolescents, as it is so in Western contexts.

### 1.7. Research Hypotheses

Previous studies have had two main limitations. First, although there have been some studies conducted in the local context [9,36,57,58,59], they failed to extensively embed the multilevel contexts regarding the individual, peer, and family factors. Second, previous studies have often involved a relatively small sample size, which may undermine their reliability [60,61]. Most respondents have been recruited from Hong Kong’s secondary schools and have been relatively young, and college students have been less involved [62]. This study builds on the existing literature by assessing the relationships between childhood abuse and trauma, peer alienation, hopelessness and depressive symptoms among college Chinese students in Hong Kong. It proposes three hypotheses:

**Hypothesis** **1.***The experience of childhood abuse and trauma is positively related to depressive symptoms in Hong Kong college students*.

**Hypothesis** **2.***The experience of peer alienation is positively related to depressive symptoms in Hong Kong college students*.

**Hypothesis** **3.***Hopelessness is positively related to depressive symptoms in Hong Kong college students*.

## 2. Methods

### 2.1. Sampling and Participants

A cross-sectional survey research design was conducted. A convenience sample comprised 786 Year 3 students studying for a vocational diploma (Grade 9 entry) and Year 1 students studying for a vocational diploma and foundation diploma (Grade 12 entry) in eight youth colleges from Hong Kong from September 2017 to January 2018. Such youth colleges operate under a vocational training institute established in the mid-2000s in Hong Kong, offering vocational training to post-secondary three and post-secondary six students. Their vocational programs can be categorized into three main subject areas: business and service, engineering, and design and information technology. Ng and Wong [63] mentioned that students in youth college, especially post-secondary three entry students, faced a variety of problems, ranging from poor academic results and deviant conduct issues, when they studied in their secondary school. Informed consent was obtained from the students and the principals of the eight youth colleges. Research ethics consent was sought from the Research Ethics Committee of the affiliated university. Questionnaires were completed by the students in class; the students typically required about 30 min to complete the whole questionnaire. Participation was completely voluntary, and students were assured that their non-participation would not affect their grades and services received from the institution.

### 2.2. Measures

#### 2.2.1. Childhood Abuse and Trauma

Childhood abuse and trauma were measured by the Chinese version of The Childhood Abuse and Trauma Scale (CATS) [28,64]. The scale consists of four subscales: the negative home environment/neglect; punishment; total abuse; and sexual abuse, but the sexual abuse subscale was not applied in this study because the study design did not cover the dimension of sexual abuse and the discomfort that replying to sexual abuse questions may trigger in respondents. Each dimension contains 6 to 13 items; e.g., ‘Did you ever seek outside help or guidance because of problems in your home?’. Participants were asked to choose a response on a five-point Likert scale, ranging from 0 (never) to 4 (always). A higher score indicates a higher level of child abuse and trauma. The internal reliability of the scale was good, with an internal consistency of 0.90 or above [28]; the scale had an internal consistency of 0.93 in this study. Its validity was supported by correlations with measurements of dissociation, difficulty in interpersonal relationships, and victimization [28]. Moreover, the Chinese version of CATS demonstrated good internal reliability and discriminant validity [64].

#### 2.2.2. Depressive Symptoms

Depressive symptoms was administered by the Chinese version of The Beck Depression Inventory-II [65]. It can be used as a depression screening instrument for non-clinical samples. The scale consists of 21 items that describe symptoms of depression. Each item has four response options and a possible score of 0 to 3; e.g., to measure sadness (item 1), the response options range from ‘I do not feel sad’ (score of 0) to ‘I am so sad and unhappy that I can’t stand it’ (score of 3). The maximum total score is 63, and a higher score indicates more depressive symptoms. The cut-off score for depressive symptoms in this study was 14, following previous studies [66]. The internal consistency (Cronbach’s alpha) was high in previous studies, ranging from 0.75 to 0.88, and the internal consistency was 0.92 in this study. Its validity was supported by correlations with measurements of suicidal ideation, peer acceptance, parental acceptance, and school performance [65].

#### 2.2.3. Peer Alienation

Peer alienation was measured by the subscale of the Inventory of Parent and Peer Attachment [47]. The scale includes three dimensions: quality of communication, mutual trust, and feelings of anger and alienation. However, only the subscale of feelings of anger and alienation was used in this study because the other dimensions of the scale were not the focus of this study. This subscale includes seven items about peer alienation; e.g., ‘Talking over my problems with friends makes me feel ashamed or foolish’ (PA). Participants were asked to choose a response on a five-point Likert scale with responses that ranged from 1 (almost never or never true) to 5 (always never or always true). Higher scores indicate a high level of peer alienation. In a recent Chinese study [67], the internal reliability (α = 0.72) of the peer alienation subscale was good; in this study, the scale had an internal consistency of 0.68. Its validity was supported by correlations with measurements of self-concept and loneliness [47].

#### 2.2.4. Hopelessness

Hopelessness was measured by The Hopelessness Subscale of the Chinese Hopelessness Scale (C-HOPE) [68]. For each of these items, the respondents were asked to answer on a four-point Likert scale ranging from 1 (strongly agree) to 4 (strongly disagree); e.g., ‘My future seems dark to me.’ Higher scores indicated higher degrees of hopelessness. According to Shek [68], the instrument had good internal consistency (α = 0.91); the internal consistency of C-HOPE was 0.90 in this study. Furthermore, C-HOPE was found to significantly correlate with other measures of positive mental health and predictors of suicidal behaviors.

### 2.3. Analytical Strategy

First, descriptive analyses were performed to analyze the socio-demographic characteristics and the prevalence of depressive symptoms in the sample (*n* = 786). Second, person correlation analyses were used to analyze the correlations of the study variables. Third, independent *t*-test and one-way (ANOVA) on depressive symptoms were conducted to compare if the study variables differed in terms of demographic variables. Fourth, multiple linear regression was conducted on the studied variables with BDI-II scores as the dependent variable.

## 3. Results

### 3.1. Socio-Demographic Data and the Prevalence of Depressive Symptoms of the Participant

The participants’ demographics are shown in Table 1. The participants consisted of 528 males (67.2%) and 258 females (32.8%). The age of the participants ranged from 18 to 21; 232 (29.5%) participants were 18 years old, 287 (36.5%) participants were 19 years old, 174 (22.1%) participants were 20 years old, and 93 (11.8%) participants were 21 years old. With reference to the Beck Depression Inventory-II (BDI-II) scores of 14 or above, a total of 354 (45%) participants had depressive moods; 86 (10.9%) participants were in their third year of the vocational diploma (Secondary 3 entry), 550 (70%) were studying for the vocational diploma in Vocation Education (Secondary 6 entry), and 150 (19.1%) were studying for the foundation diploma. Only 138 (17.5%) participants were diagnosed with special educational needs. A total of 556 (70.7%) participants described their parents as married, 18 (2.3%) stated that their parents were remarried, 126 (16%) had divorced parents, 22 (2.8%) had separated parents, and 33 (4.25) had cohabiting parents. The distribution of the participants’ income was as follows: 163 (20.7%) with HKD5,000 to 10,000 per month, 437 (55.6%) with HKD10,001 to 30,000 per month, 131 (16.7%) with HKD30,001 to 60,000 per month, and 28 (3.6%) with HKD60,001 or above per month. We found significant association of depressive symptoms with different education programmes and the special education needs status of students.

### 3.2. Correlation Analysis of Variables in the Participants

As shown in Table 2, all of the correlations were similar to what was predicted. All studied variables, including child abuse and trauma (*r* = 0.399), peer alienation (*r* = 0.218), and hopelessness (*r* = 0.485) were significantly and positively associated with depressive symptoms. Thus, adolescents with more peer alienation, hopelessness, and child abuse and trauma had higher depressive symptoms.

### 3.3. Results of Independent t-Test and ANOVA on Depressive Symptoms

As shown in Table 3, by comparing the means using independent *t*-test and ANOVA, the students with SEN in the eight colleges presented higher depressive symptoms scores (*t* = −3.57, *p* < 0.001). Several studies have shown that students with SEN felt less socially integrated and more often segregated compared to their peers without SEN. Moreover, students with SEN, on average, also had fewer friends [69]. Furthermore, students with SEN also displayed more loneliness than students without SEN [70]. To conclude, previous studies have shown that people who are less socially integrated, have fewer friends, and show more loneliness, are more vulnerable to developing depressive moods [71]. Thus, students with SEN have significantly higher depression scores than those without SEN. However, there are no significant differences between the students in respect of gender, years of study, family income, and so on.

### 3.4. Regression

As indicated in Table 4, a multiple linear regression was conducted to predict college students’ depressive symptoms based on their risk factors. A satisfactory regression equation was found, F (3, 781) = 124.126, *p* < 0.001, with an *R*^2^ of 0.32. Childhood abuse and trauma, peer alienation, and hopelessness were thus significant predictors of depressive symptoms in adolescents.

## 4. Discussion

This study finds a positive relationship between childhood abuse and trauma and depressive symptoms among Chinese college students, supporting Hypothesis 1. The result is consistent with previous studies, that childhood trauma is significantly associated with depressive symptoms in Chinese adolescents [72], and Western findings that childhood abuse is significantly related to adolescent’s depressive symptoms [30].

The result may have several explanations. First, experiencing childhood abuse and trauma may impair a child’s development [73]. Karakuş [74] mentioned that abuse and neglect may affect individuals’ self-perception and lead to low self-esteem [74]. Individuals with low self-esteem are likely to display risky behaviors and undergo negative development [75]. Abused children who have frequently experienced criticism, rejection, and control at the hands of significant others tend to form a negative self-perception, which reinforces a tendency towards depression [76].

Second, based on the cognitive model of depression [34], maladaptive cognitive processes and information processes are the products of earlier negative affective experiences characteristic of depression. The experience of childhood abuse is likely to have a particularly negative impact on individuals’ later schema and lead to a negative belief system. This may also show lasting impacts on an individual’s response to stressors given the cognition of an individual may be affected by the early encoded experience with its effects on neural development and differentiation [77], which in turn may cause depression [33]. Likewise, according to attachment theory [32], significant others are essential to the development of an individual’s representational or internal working models of the world. Individuals who had an abusive childhood tend to develop a negative schema of themselves and others, which may in turn increase their likelihood of depression [78]. Hence, this study finds that childhood abuse and trauma are positively related to depressive symptoms in Hong Kong college students.

This study indicates a positive relationship between peer alienation and depressive symptoms among college participants, giving support to Hypothesis 2. This finding is consistent with past studies, showing that peer alienation is significantly correlated with depression [51] and that children who are alienated from their peer groups are at higher risk of depressive symptoms [79]. A developmental perspective may explain the positive relationship between peer alienation and depressive symptoms in college students. During adolescence, adolescents tend to explore wider social environments. They shift their attention and relationships from family to peer groups in their school and community [40]. The external environment (peer factors) is highly significant to their development [42]. Therefore, it can be inferred that individuals who feel alienated from their peers avoid developing positive interpersonal relationships with them. This is consistent with previous research, showing that peer alienation tends to be an obstacle for adolescents to make friends, which might result in a smaller social network, poor social skills, and a lack of self-confidence, and these factors are positively associated with depression and other psychological symptoms [46]. Hence, this study finds that peer alienation is positively related to depressive symptoms in Hong Kong college students.

This study shows a positive relationship between hopelessness and depressive symptoms among college students, supporting Hypothesis 3. This finding is in line with previous studies, preforming that hopelessness is a proximal risk factor for depression [56]. Empirical studies have demonstrated that hopelessness is correlated with depressive disorders [80] and significantly predicts increased depressive symptoms [81]. According to Beck [35], hopelessness is a set of negative feelings and expectations that people have about themselves and their future, in addition to inadequate motivation. In this study, it can be inferred that adolescents who feel hopeless tend to have a negative perspective on current and future events. Adolescents without hope also tend to have difficulties in solving problems [82]. When individuals who feel despaired and hopeless become overwhelmed, their thoughts are likely to become increasingly negative [52], which in turn leads to depressive symptoms [83]. Hence, hopeless people may experience negative thinking and be deficient in problem solving, which may contribute to depressive symptoms.

## 5. Implications

This study supported an ecological model and developmental psychopathology theory of depressive symptoms for Chinese college students in Hong Kong. The findings are particularly important or have implications on depressive symptoms in adolescents in Chinese society. First, this study identifies several key factors (peer alienation, hopelessness, and childhood abuse and trauma) that contribute to the development of depressive symptoms in adolescents in Hong Kong. Second, it is a study on the effects of risk factors of depressive symptoms on the relationships between ontogenic systems (hopelessness) and the microsystem (childhood abuse and trauma, and peer alienation) by using an ecological model and developmental psychopathology theory.

The current study revealed likely reasons for individuals’ increased susceptibility to depressive symptoms among adolescents with childhood abuse and trauma experiences. These reasons include peer alienation and hopelessness. Various interventions can be used to undermine the tendency for depressive symptoms or the likelihood of engaging in depressive symptoms among abused adolescents, namely, strengthening peer relationships, reducing peer alimentation, reducing hopelessness, and reducing childhood abuse and trauma. It is recommended to consider these contextual factors for the care of depression in adolescents in Hong Kong. Social workers may apply this knowledge to design prevention and early intervention programs. Some researchers have proposed using multicomponent and multilevel interventions to address the multiple risk factors related to the development and maintenance of depression [84]. Such kinds of interventions emphasize factors that may lead to depressive symptoms due to their accessibility, and thus amenability to change.

To reduce childhood abuse and trauma, previous research has proposed introducing prevention at different levels, including primary, secondary, and tertiary prevention [11]. Primary prevention involves providing positive knowledge, skills, and attitudes toward parenting to build healthy families. School social workers can organize parenting talks or workshops for all parents at the first stage to identify parents who need help, and then apply the group work approach to the identified parents at the next stage. School social workers can refer parents, with their consent, to family social workers to handle the more complicated family issues. Secondary prevention aims at identifying students who have experienced childhood abuse and trauma. Relevant staff development and training programs are suggested to enhance teachers’ awareness of the signs of childhood abuse and trauma. Such programs should cover how to identify the signs and symptoms of abused students. Group work could be applied to vulnerable parents. Tertiary prevention is intended to prevent additional harm from being done to a child after child abuse is established, this is the main role of a social worker. Currently, social workers are requested to follow the Social Welfare Department’s procedural guide on abuse cases in Hong Kong [85]. This guide covers basic information on child abuse, how to handle inquiries and referrals, how to investigate suspected cases, and the roles of relevant departments and organizations.

In addition, a number of steps can be taken to reduce peer alienation and strengthen peer relationships. First, school social workers can be more proactive in designing intervention programs, such as socialization weeks or tutoring activities, to reduce students’ sense of alienation. Moreover, when students are more engaged in school activities, they can better understand their school environment, obligations, and rights, which increase their sense of control and in turn decrease their sense of loneliness [86]. Second, research has found that peer alienation has a significant association with peer trust and self-efficacy [87]. Therefore, school social workers can provide opportunities for students to build connections and a sense of community. For example, they can do so by conducting outreach related to peer communications in the first phase, and then conducting large-group outreach sessions in the next phase. Not only can it help students to experience a positive affect in their social interactions and develop healthy peer relationships, but also it can teach them how to communicate in ways that help them to regain a sense of self-efficacy, which can in turn reduce the sense of peer alienation. Thus, it would be beneficial to introduce appropriate psychoeducation that integrates the relation between the constructs (e.g., maltreatment history, peer alienation, sense of hopelessness, and depressive symptoms) to the treatment process, because it can improve receptivity by adolescents and help in overcoming potential resistance to the treatment. Moreover, considering the adolescent’s attachment style and his/her current problems may assist the therapist in building a better therapeutic relationship, which, in turn, can contribute to effective treatment goals with adolescents [88].

Furthermore, school social workers could provide students with cognitive training to reduce hopelessness. Cognitive training boosts people’s memory and specifically reduces rumination and lessens the feeling of hopelessness. For instance, it has been reported that depressed inpatients who participated in ‘Memory Specificity Training’ (MEST) showed a notable reduction in hopelessness [89]. Moreover, Marchetti, Alloy and Koster [90] suggested that a Positive Psychology Intervention, using the Best Possible Self technique (BPS) [91], could be a promising intervention for hopeless individuals to reduce their levels of hopelessness. The BPS technique requires the participant to imagine and write down about their best possible self in terms of feelings, thoughts, and situations in the future where they have reached everything desired. The results reported that a single-session BPS intervention could lead to a significant enhancement in expectations of a favourable future and a decrease in negative scenarios, followed by mood change [92,93]. Moreover, it has been proved that BPS intervention can lower the levels of hopelessness among suicidal patients [94].

## 6. Limitations

First, a cross-sectional design was used in this study, so it was not possible to attribute its results that childhood abuse and trauma, peer alienation, and hopelessness contribute to depressive symptoms. Further studies can test the long-term impacts of childhood abuse and trauma, peer alienation, and hopelessness on depressive symptoms by adopting longitudinal studies.

Second, the reports of childhood abuse and trauma collected in this study were retrospective self-reports. These self-reports were not validated by other sources, so they might have been affected by recall bias. However, it has been demonstrated that CATS, the measure chosen for this study, has excellent psychometric properties, including strong convergent validity with therapists’ ratings of abuse. Similarly, other proposed correlates, including peer alienation, hopelessness, and depressive symptoms, were also self-reported. Therefore, the associations among these variables might have been overestimated. However, all of the measurements are validated and commonly used tools for studies of adolescents. Data on adolescent depression, childhood abuse and trauma experience, peer alienation, and hopelessness were not collected from multiple informants, which would have provided a clearer picture. However, many studies have shown that researchers can count on the validity and reliability of adolescents as informants about their depressive conditions, whereas other informants, such as parents, may cover up the internalizing symptoms of adolescents [95], which can significantly undermine individual differences. In addition, because teachers and parents are not likely to be familiar with how adolescents and adolescents make social interactions with their peers (e.g., peer alienation), self-reports from adolescents provide essential data on them. Studies have also indicated the value of assessing an individual’s perception of their internal states (e.g., hopelessness), which may not be easily observable by external parties [96]. In addition, using self-reports was relevant to this study’s examination of the links between childhood experience, adolescent social life, cognitive style, and depressive symptoms.

Third, the convenience sample in this study might not be an accurate representation of the general student population in Hong Kong. A larger sample and more variables are required; future studies could examine students attending different colleges and universities, as well as different age groups [97].

Fourth, to understand the long-term effects of childhood abuse and trauma, it is essential to do more than investigate the direct effects of childhood abuse and trauma. It is important to examine the experiences of adolescents with childhood abuse and trauma—for example, their experiences of peer alienation and hopelessness. Moreover, because there may be various mediators in the linkage between childhood abuse and trauma and depressive symptoms, more analyses are needed to examine the indirect effects by studying multiple mediators in parallel or serially.

## 7. Conclusions

The ontogenic system and microsystems of the ecological model and developmental psychopathology theory were applied as a theoretical framework to understand Hong Kong college students’ depression. Individual, peer, and family correlates were integrated to examine college student depression in Hong Kong, and childhood abuse and trauma, peer alienation, and hopelessness were demonstrated to be significantly and positively associated with depressive symptoms in college students in Hong Kong.

## Figures and Tables

**Table 1 ijerph-20-04304-t001:** Socio-demographic data and the prevalence of depressive symptoms in the sample of Hong Kong college students of Study Participants (*n* = 786).

Variables	Total	With Depressive Symptoms	Without Depressive Symptoms	X2	Df	*p*
Gender				0.293	1	0.588
Male	528	240	288			
Female	258	112	146			
Age				3.015	3	0.388
18 years old	232	94	138			
19 years old	287	135	152			
20 years old	174	93	91			
21 years old	93	40	53			
Education				7.728	2	0.021
Diploma in Vocation Education (DVE) (Secondary 3 entry)	86	50	36			
Diploma in Vocation Education (DVE) (Secondary Six entry)	550	242	308			
Foundation Diploma (DFS) (Secondary Six entry)	150	60	90			
Special educational needs				7.356	1	0.007
With	64	39	25			
Without	713	309	404			
Parents’ marital status				4.9	5	0.428
Married	556	247	309			
Remarried	18	9	9			
Divorced	126	60	66			
Separated	22	7	15			
Cohabiting	33	12	21			
Other	22	13	9			
Distribution of household income				3.370	4	0.498
HKD5000 to 10,000 per month	163	80	83			
HKD10,001 to 30,000 per month	437	188	249			
HKD30,001 to 60,000	131	63	68			
HKD60,001 or above per month	28	10	18			

**Table 2 ijerph-20-04304-t002:** Correlation Matrix of Study Variables (*n* = 786).

Variables		Mean/(SD)	1	2	3	4
1.	Hopelessness	2.32/(0.58)	1			
2.	Peer Alienation	2.83/(0.60)	0.190 **	1		
3.	Child abuse and trauma	1.40/(0.66)	0.266 **	0.260 **	1	
4.	Depressive symptoms	0.69/(0.52)	0.485 **	0.218 **	0.399 **	1

Notes: ** Correlation is significant at the *p* < 0.01 level.

**Table 3 ijerph-20-04304-t003:** Result of Independent *t*-test and ANOVA on depressive symptoms.

Groups	*n*	Mean (SD)	t/F	*p*-Value
Gender				
Male	528	0.69 (0.53)	*t* = 0.53	0.60
Female	258	0.67 (0.51)		
Birthplace				
Hong Kong	552	0.70 (0.54)	*t* = 1.46	0.15
China	229	0.64 (0.49)		
Religion				
Without religion	621	0.67 (0.50)	*t* = −1.73	0.09
With religion	164	0.74 (0.59)		
SEN				
With SEN	713	0.66 (0.51)	*t* = −3.57	0.00
Without SEN	64	0.91 (0.59)		
Marital Status				
Married	574	0.68 (0.53)	*t* = 0.21	0.35
Divorced	203	0.67 (0.50)		
Salary				
$30,000 or below	600	0.69 (0.51)	*t* = 0.14	0.28
$30,001 or above	159	0.68 (0.54)		
Course				
DVE (Entry 3)	86	0.79 (0.52)	F = 2.72	0.07
DVE (Entry 6)	550	0.68 (0.53)		
DFS	150	0.50 (0.50)		

**Table 4 ijerph-20-04304-t004:** Multiple linear regression in predicting depressive symptoms (BDI-II) (*n* = 786).

Predictor	B	SE	*t*	*p*
Child abuse and trauma	0.184	0.020	9.048	0.000
Hopelessness	0.357	0.028	12.905	0.000
Peer Alienation	0.061	0.027	2.264	0.024

## Data Availability

The data presented in this study are available on request for the corresponding author.

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
