# Peer review of "Individual, Peer, and Family Correlates of Depressive Symptoms among College Students in Hong Kong"

_ijerph, 2023, doi:10.3390/ijerph20054304_

Round 1
Reviewer 1 Report
The study aims to investigate a relationship between depressive symptoms and childhood abuse and trauma, peer alienation and hopelessness in a sample of Hong Kong college students.
The topic is certainly interesting because, as the authors claim, there is a very high incidence of depression among young people in Hong Kong. It is therefore appreciable that researchers are dedicated to studying this phenomenon to try to curb it. However, no new elements seem to emerge from this study that would allow for a deeper reading of the phenomenon. It is simply a cross-sectional l study that relates variables already known in the literature to be correlated. Unfortunately, these analyses do not tell us much about the predictive effect of these variables, only that there is a relationship between them. Alienation could lead to depressive symptoms, but at the same time depression could lead to alienation from peers. I believe more effort should be made to convey the innovative contribution of this study.
If, as the authors claim, depression is such a serious problem in Hong Kong, they should first devote part of the introduction to try to explain what elements (individual, relational and social) contribute to generating such high levels of depression among young people.
Also missing is reference to the period in which the study was conducted: pre or post Covid-19? Is there a relationship between the incidence of these symptoms and the pandemic?
In my opinion, it might benefit the article to refer to the literature on Adverse Childhood Experiences (ACEs) which broadens the perspective beyond abuse. In addition, when the authors talk about alienation, it might also be useful to talk about the importance of social (peer) support, including references to relevant literature.
In general, in my opinion, a review of the bibliographical references, which are mostly dated, would be needed. It is fine to start with classic authors such as Bronfenbrenner, Beck and Bowlby, but I believe that several studies have been conducted on these issues, applied to more contemporary contexts.
I would then specify better what kind of abuse they measured with their scale, if they excluded sexual abuse. Is it really abuse or neglect?
On p. 5, line 176 there is a typo: Each should be capitalized.
Finally, how do they think the results of this survey can be used from an application perspective? It is clear that negative phenomena such as abuse and alienation can contribute to sustaining the development of depressive symptoms (this is not new), so what does their model propose that would help social workers deal with this problem?
I believe that only by addressing these critical issues can the paper be considered for publication in the journal.
Author Response
My response to Reviewer A
The study aims to investigate a relationship between depressive symptoms and childhood abuse and trauma, peer alienation and hopelessness in a sample of Hong Kong college students.
- The topic is certainly interesting because, as the authors claim, there is a very high incidence of depression among young people in Hong Kong. It is therefore appreciable that researchers are dedicated to studying this phenomenon to try to curb it. However, no new elements seem to emerge from this study that would allow for a deeper reading of the phenomenon. It is simply a cross-sectional l study that relates variables already known in the literature to be correlated. Unfortunately, these analyses do not tell us much about the predictive effect of these variables, only that there is a relationship between them. Alienation could lead to depressive symptoms, but at the same time depression could lead to alienation from peers. I believe more effort should be made to convey the innovative contribution of this study.
- Descriptive analyses were performed to analyze the socio-demographic characteristics and the prevalence of depressive symptoms of the sample (N=786). Second, person correlation analyses were used to analyse the correlations of the study variables. Third, independent T-test and one-way (ANOVA) on depressive symptoms were conducted to compare if the study variables differ in the demographic variables. Fourth, multiple linear regression was conducted on the studied variables with BDI-II scores as the dependent variable.
- If, as the authors claim, depression is such a serious problem in Hong Kong, they should first devote part of the introduction to try to explain what elements (individual, relational and social) contribute to generating such high levels of depression among young people.
- A paragraph on Individual, Family, and Peer Risk correlates of depression among Chinese adolescents was supplemented
- Also missing is reference to the period in which the study was conducted: pre or post Covid-19? Is there a relationship between the incidence of these symptoms and the pandemic?
- The study was conducted from September 2017- January 2018; therefore, there is no relationship between the incidence of these symptoms and the pandemic.
- In my opinion, it might benefit the article to refer to the literature on Adverse Childhood Experiences (ACEs) which broadens the perspective beyond abuse. In addition, when the authors talk about alienation, it might also be useful to talk about the importance of social (peer) support, including references to relevant literature.
- Some literature on the importance of social (peer) is supplemented. For example, Dishion & Tipsord, 2011; Lam et al., 2014; Eccles & Roeser, 2009
- In general, in my opinion, a review of the bibliographical references, which are mostly dated, would be needed. It is fine to start with classic authors such as Bronfenbrenner, Beck and Bowlby, but I believe that several studies have been conducted on these issues, applied to more contemporary contexts.
- Yes, some updates were added.
- I would then specify better what kind of abuse they measured with their scale, if they excluded sexual abuse. Is it really abuse or neglect?
- The scale consists of four subscales: the negative home environment/neglect; punishment; total abuse and sexual abuse. In this case, although sexual abuse is excluded, other subscales measured the respondents' childhood experiences of abuse and neglect.
- On p. 5, line 176 there is a typo: Each should be capitalized.
- Yes, revised.
- Finally, how do they think the results of this survey can be used from an application perspective? It is clear that negative phenomena such as abuse and alienation can contribute to sustaining the development of depressive symptoms (this is not new), so what does their model propose that would help social workers deal with this problem?
- The current study has found that some key factors (peer alienation, hopelessness) contribute to the association between childhood abuse and neglect and trauma, and depressive symptoms in Chinese adolescents. In other words, adolescents who have childhood abuse and trauma experiences are likely to alienate themselves from peers and develop a feeling of hopelessness, contributing to depressive symptoms. Therefore, social workers should pay attention to reducing the impact of depressive symptoms by reducing peer alienation, strengthening peer relationships, and building hope.

Reviewer 2 Report
The authors examined the Individual, Peer, and Family Correlates of Depressive Symptoms among college students in Hong Kong. My comments are:
1) Which variable is the family correlate? Explain more why it is a family correlate.
2) Introduction: How "a representative study showed that a total of 9,518 secondary school students, (38.36% males and 46.13% females), had a moderate to severe level of depression (The Center for Epidemiological Studies-Depression (CES-D) ≥ 21) (Wu et al., 2016)." relates to Young Adults?
3) 1.2. Conceptual Framework of the Study: (a) Kindly provide a figure to better illustrate the framework. (b) Is this a theoretical framework as mentioned in the conclusion?
4) What is the problem statement?
5) 1.6. Research Hypotheses: (a) For the first limitation, why is it a limitation if other studies investigate the individual, peer, and family factors separately? (b) Please cite the "previous studies" for the second limitation.
6) Table 1: Needs some alignment formatting on the characteristics so that it's easier to read.
7) 2.2.1. Childhood Abuse and Trauma: Any reference to the Chinese version?
8) Table 2: No need to mention "*Correlation is significant at p < 0.05."
9) 6. Limitations: (a) Please provide suggestions for future research for each limitation. (b) The third limitation is on adolescents, not young adults???
Author Response
My response to Reviewer B
1) Which variable is the family correlate? Explain more why it is a family correlate.
- Childhood abuse and trauma is the family correlate. For the explanation, please refer to lines 150-157.
2) Introduction: How "a representative study showed that a total of 9,518 secondary school students, (38.36% males and 46.13% females), had a moderate to severe level of depression (The Center for Epidemiological Studies-Depression (CES-D) ≥ 21) (Wu et al., 2016)." relates to Young Adults?
- Yes, it relates to adolescents. The term “young adult” was misused, and it was revised.
3) 1.2. Conceptual Framework of the Study: (a) Kindly provide a figure to better illustrate the framework. (b) Is this a theoretical framework as mentioned in the conclusion?
- The conceptual framework was revised. Please refer to lines 100-148.
4) What is the problem statement?
- This study aimed to investigate the personal, family, and peer correlates of depressive symptoms among adolescents in Hong Kong. Specifically, this study investigated the contributions of hopelessness, childhood abuse and neglect, and peer alienation to depressive symptoms in samples of adolescents in Hong Kong.
5) 1.6. Research Hypotheses: (a) For the first limitation, why is it a limitation if other studies investigate the individual, peer, and family factors separately? (b) Please cite the "previous studies" for the second limitation.
- It is because they have failed to extensively embed the multilevel contexts regarding the individual, peer, and family factors
- Kwok & Gu, 2017; Leung et al., 2017 are added to support second limitation
6) Table 1: Needs some alignment formatting on the characteristics so that it's easier to read.
- Revised
7) 2.2.1. Childhood Abuse and Trauma: Any reference to the Chinese version?
- Yes, He et al., 2008 added
8) Table 2: No need to mention "*Correlation is significant at p < 0.05."
- Revised
9) 6. Limitations: (a) Please provide suggestions for future research for each limitation. (b) The third limitation is on adolescents, not young adults???
- Revised, please refer to the "limitation part"

Round 2
Reviewer 1 Report
I think the authors have done a lot of work to improve the article and that the paper is now ready to be published